# How Low Can We Go: Trading Memory for Error in Low-Precision Training

**Chengrun Yang\*, Ziyang Wu\*, Jerry Chee, Christopher De Sa, Madeleine Udell**
Cornell University
{cy438,zw287,jc3464,cmd353,udell}@cornell.edu

## ABSTRACT

Low-precision arithmetic trains deep learning models using less energy, less memory and less time. However, we pay a price for the savings: lower precision may yield larger round-off error and hence larger prediction error. As applications proliferate, users must choose which precision to use to train a new model, and chip manufacturers must decide which precisions to manufacture. We view these precision choices as a hyperparameter tuning problem, and borrow ideas from meta-learning to learn the tradeoff between memory and error. In this paper, we introduce Pareto Estimation to Pick the Perfect Precision (PEPPP). We use matrix factorization to find non-dominated configurations (the Pareto frontier) with a limited number of network evaluations. For any given memory budget, the precision that minimizes error is a point on this frontier. Practitioners can use the frontier to trade memory for error and choose the best precision for their goals.

## 1 INTRODUCTION

Training modern-day neural networks is becoming increasingly expensive as task and model sizes increase. The energy consumption of the corresponding computation has increased alarmingly, and raises doubts about the sustainability of modern training practices (Schwartz et al., 2019). Low-precision training can reduce consumption of both computation (De Sa et al., 2018) and memory (Sohoni et al., 2019a), thus minimizing the cost and energy to train larger models and making deep learning more accessible to resource-limited users.

Low-precision training replaces 32-bit or 64-bit floating point numbers with fixed or floating point numbers that allocate fewer bits for the activations, optimizers, and weights. A rich variety of methods appear in recent literature, including bit-centering (De Sa et al., 2018), loss scaling (Micikevicius et al., 2017) and mixed-precision training (Zhou et al., 2016). There is however a fundamental tradeoff: lowering the number of bits of precision increases the *quantization error*, which may disrupt convergence and increase downstream error (Zhang et al., 2016; Courbariaux et al., 2014; Gupta et al., 2015). The *low-precision configuration* is thus a hyperparameter to be chosen according to the resource constraints of the practitioner.

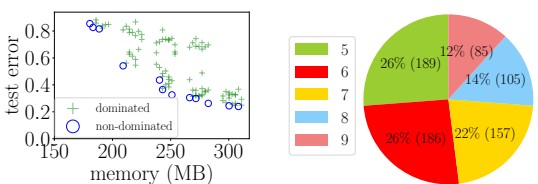

(a) error vs memory on CIFAR-10 (Krizhevsky et al., 2009)

(b) # activation bits in non-dominated configurations

Figure 1: Test error vs memory for ResNet-18 across 99 low-precision floating point configurations. Figure (a) shows the tradeoff on CIFAR-10. (Non-dominated points are blue circles.) Figure (b) shows that the best precision to use varies depending on the memory budget, on 87 image datasets. See Section 4 for experimental details.

How should we choose this hyperparameter? Many believe the highest allowable precision (given a memory budget) generally produces the lowest error model. However, there are typically many ways

---

\*Equal contribution.

to use the same amount of memory. As shown in Figure 1(a), some of these configurations produce much lower error than others! Figure 1(b) shows that no one configuration dominates all others. We might consult previous literature to choose a precision; but this approach fails for new applications.

**Our goal is to efficiently pick the best low-precision configuration under a memory budget.** Efficiency is especially important for resource-constrained practitioners, such as individual users or early-stage startups. To promote efficiency, we use a meta-learning (Lemke et al., 2015; Vanschoren, 2018; Hospedales et al., 2020) approach: we train a small number of cheap very-low-precision models on the dataset to choose the perfect precision. The gains from choosing the right low-precision format can offset the cost of this extra training — but each precision must be chosen carefully to realize the benefits of low-precision training.

We use ideas from multi-objective optimization to characterize the tradeoff between memory and error and identify the *Pareto frontier*: the set of non-dominated solutions. Users will want to understand the tradeoff between error and memory so they can determine the resources needed to adequately train a model for a given machine learning task. This tradeoff may also influence the design of application-specific low-precision hardware, with profound implications for the future (Hooker, 2020). For example, among all 99 low-precision configurations we tried, we identified some configurations that are Pareto optimal across many different tasks (listed in Appendix D). These results could help hardware manufacturers decide which precision settings are worth manufacturing.

Computing the Pareto frontier by training models for all low-precision configurations is expensive and unnecessary. Cloud computing platforms like Google Cloud and Amazon EC2 charge more for machines with the same CPU and double memory: 25% more on Google Cloud[*] and 100% more on Amazon EC2[*] as of mid-September 2021. We use techniques from meta-learning to leverage the information from other low-precision training runs on related datasets. This approach allows us to *estimate* the Pareto frontier without evaluating all of the low-precision configurations.

Our system, Pareto Estimation to Pick the Perfect Precision (PEPPP), has two goals. The first goal is to find the Pareto frontiers of a collection of related tasks, and is called *meta-training* in the meta-learning literature. Meta-training requires a set of *measurements*, each collected by training and testing a neural network with a given precision on a dataset. This information can be gathered offline with a relatively large resource budget, or by crowdsourcing amongst the academic or open-source community. Still, it is absurdly expensive to exhaustively evaluate all measurements: that is, every possible low-precision configuration on every task. Instead, we study how to choose a subset of the possible measurements to achieve the best estimate. The second goal we call *meta-test*: using the information learned on previous tasks, how can we transfer that information to a new task to efficiently estimate its Pareto frontier? This goal corresponds to a resource-constrained individual or startup who wants to determine the best low-precision configuration for a new dataset.

Both meta-training and meta-test rely on matrix completion and active learning techniques to avoid exhaustive search: we make a subset of all possible measurements and predict the rest. We then estimate the Pareto frontier to help the user make an informed choice of the best configuration. We consider two sampling schemes: uniform and non-uniform sampling. Uniform sampling is straightforward. Non-uniform sampling estimates the Pareto frontier more efficiently by making fewer or even no high-memory measurements.

To the best of our knowledge, PEPPP is the first to study the error-memory tradeoff in low-precision training and inference without exhaustive search. Some previous works show the benefit of low-precision arithmetic in significant energy reduction at the cost of a small accuracy decrease (Hashemi et al., 2017), and propose hardware-software codesign frameworks to select the desired model (Langroudi et al., 2019). Many other papers focus on low-precision inference only and use it for model compression (Wu et al., 2018; Cai & Vasconcelos, 2020). Another line of work seeks state-of-the-art (SOTA) performance with low-precision training, using carefully designed learning schedules (Sun et al., 2019; 2020). Our work imagines low-precision as a way to **reduce training or deployment costs of customized models** trained on proprietary datasets (not SOTA models on ML benchmarks). Even so, we show in Section 4 that our work picks promising models for every memory budget, which enables near-SOTA results on CIFAR-10.

---

[*]https://cloud.google.com/ai-platform/training/pricing
[*]https://aws.amazon.com/ec2/pricing/on-demand

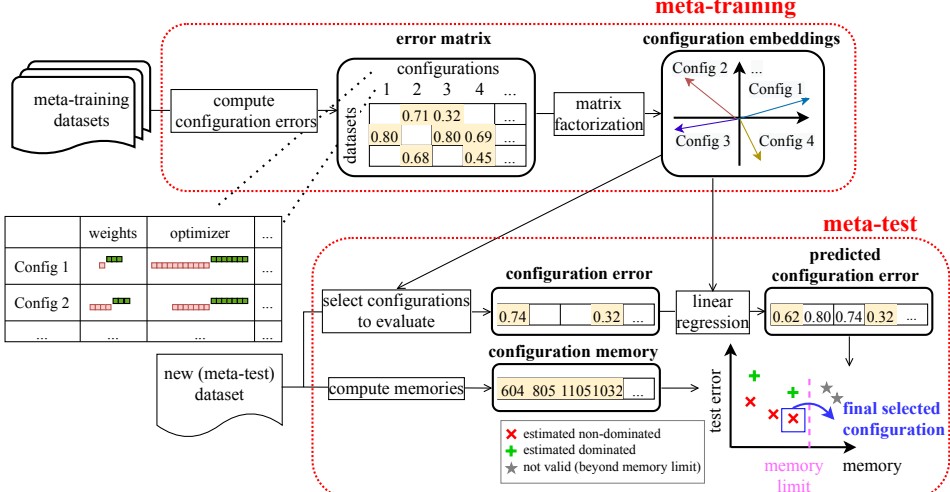

Figure 2: The PEPPP workflow. We begin with a collection of (meta-) training datasets and low precision configurations. In the meta-training phase, we sample dataset-configuration pairs to train, and compute the misclassification error. We use matrix factorization to compute a low dimensional embedding of every configuration. In the meta-test phase, our goal is to pick the perfect precision (within our memory budget) for the meta-test dataset. We compute the memory required for each configuration, and we select a subset of fast, informative configurations to evaluate. By regressing the errors of these configurations on the configuration embeddings, we find an embedding for the meta-test dataset, which we use to predict the error of every other configuration (including more expensive ones) and select the best subject to our memory budget.

Figure 2 shows a flowchart of PEPPP. The rest of this paper is organized as follows. Section 2 introduces notation and terminology. Section 3 describes the main ideas we use to actively sample configurations and approximate Pareto frontiers. Section 4 shows experimental results.

## 2 NOTATION AND TERMINOLOGY

**Low-precision formats.** We use floating point numbers for low-precision training in this paper. As an example, Figure 3 shows an 8-bit floating point number with 3 exponent bits and 4 mantissa bits. A specific low-precision representation with a certain number of bits for each part is called a *low-precision format*. We may use different low-precision formats for the weights, activations, and optimizer. In the case of using two formats to train and represent a

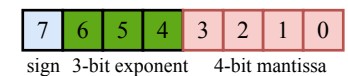

Figure 3: An 8-bit floating point number representing $(-1)^{\text{sign}} \cdot 2^{\text{exponent}-7} \cdot 1.b_3b_2b_1b_0$.

neural network (as discussed in Section 3.1), these two formats are called *Format A* and *B*. A specific combination of these two formats is a hyperparameter setting of the neural network, and we call it a *low-precision configuration*.

**Math basics.** We define $[n] = \{1, \ldots, n\}$ for a positive integer $n$. With a Boolean variable $\mathcal{X}$, the indicator function $\mathbb{1}(\mathcal{X})$ equals 1 if $\mathcal{X}$ is true, and 0 otherwise. With a scalar variable $x$, we use $x_+$ to denote $\max\{x, 0\}$. $\subseteq$ and $\subset$ denote subset and strict subset, respectively.

**Linear algebra.** We denote vector and matrix variables respectively by lowercase letters ($a$) and capital letters ($A$). All vectors are column vectors. The Euclidean norm of a vector $a \in \mathbb{R}^n$ is $\|a\| := \sqrt{\sum_{i=1}^{n} a_i^2}$. To denote a part of a matrix $A \in \mathbb{R}^{n \times d}$, we use a colon to denote the varying dimension: $A_{i,:}$ and $A_{:,j}$ (or $a_j$) denote the $i$th row and $j$th column of $A$, respectively, and $A_{ij}$ denotes the $(i, j)$-th entry. With an ordered index set $\mathcal{S} = \{s_1, \ldots, s_k\}$ where $s_1 < \ldots < s_k \in [d]$, we denote $A_{:\mathcal{S}} := [A_{:,s_1} \quad \cdots \quad A_{:,s_k}] \in \mathbb{R}^{n \times k}$. Given two vectors $x, y \in \mathbb{R}^n$, $x \preceq y$ means $x_i \leq y_i$

for each $i \in [n]$. Given a matrix $A \in \mathbb{R}^{n \times d}$ and a set of observed indices as $\Omega \subseteq [n] \times [d]$, the partially observed matrix $P_\Omega(A) \in \mathbb{R}^{n \times d}$ has entries $(P_\Omega(A))_{ij} = A_{ij}$ if $(i, j) \in \Omega$, and 0 otherwise.

**Pareto frontier.** Multi-objective optimization simultaneously minimizes $n$ costs $\{c_i\}_{i \in [n]}$. A feasible point $c^{(1)} = (c_1^{(1)}, \ldots, c_n^{(1)})$ is *Pareto optimal* if for any other feasible point $c^{(2)} = (c_1^{(2)}, \ldots, c_n^{(2)})$, $c^{(2)} \preceq c^{(1)}$ implies $c^{(2)} = c^{(1)}$ (Boyd & Vandenberghe, 2004). The set of Pareto optimal points is the *Pareto frontier*.

**Tasks, datasets, measurements and evaluations.** A *task* carries out a process (classification, regression, image segmentation, etc.) on a *dataset*. Given a deep learning model and a dataset, the training and testing of the model on the dataset is called a *measurement*. In our low-precision context, we *evaluate* a configuration on a dataset to make a measurement.

**Meta-learning.** Meta-learning transfers knowledge from past (*meta-training*) tasks to better understand the new (*meta-test*) task. To evaluate the error of meta-training, we use *meta-leave-one-out cross-validation* (meta-LOOCV): on a collection of tasks, each time we treat one task as meta-test and the others as meta-training, and average the results over all splits.

**Error matrix and memory matrix.** Given a neural network, the errors of different low-precision configurations on meta-training datasets form an *error matrix*, whose $(i, j)$-th entry $E_{ij}$ is the test error of the $j$th configuration on the $i$th dataset. To compute the error matrix, we split the $i$-th dataset into training and test subsets (or use a given split), train the neural network at the $j$-th configuration on the training subset, and evaluate the test error on the test subset. The memory required for each measurement forms a *memory matrix* $M$, which has the same shape as the corresponding error matrix. Example error and memory matrices are shown in Figure 4.

| | low-precision configurations | | | | | low-precision configurations | | | |
|---|---|---|---|---|---|---|---|---|---|
| | a | b | c | d | | a | b | c | d |
| 1 | 0.85 | 0.71 | 0.32 | 0.28 | 1 | 181 | 269 | 272 | 295 |
| 2 | 0.80 | 0.80 | 0.80 | 0.69 | 2 | 180 | 269 | 272 | 295 |
| 3 | 0.80 | 0.68 | 0.46 | 0.45 | 3 | 590 | 924 | 927 | 1032 |
| | (a) error matrix $E$ | | | | | (b) memory matrix $M$ (MB) | | | |

Figure 4: Example error and memory matrices for some datasets and low-precision configurations. Dataset 1: CIFAR-10, 2: CIFAR-100 (fruit and vegetables), 3: ImageNet-stick (Deng et al., 2009). Configuration Format A (exponent bits, mantissa bits), Format B (exponent bits, mantissa bits). a: (3, 1), (6, 7); b: (3, 4), (7, 7); c: (4, 3), (8, 7); d: (5, 3), (6, 7).

## 3 METHODOLOGY

PEPPP operates in two phases: meta-training and meta-test. First in meta-training, we learn configuration embeddings by training a neural network of a specific architecture at different low-precision configurations on different datasets. As listed in Appendix A, we study image classification tasks and a range of low-precision formats that vary in the number of bits for the exponent and mantissa for the activations, optimizer, and weights. Our hope is to avoid enumerating every possible configuration since exhaustive search is prohibitive in practice. To this end, we make a few measurements and then predict the other test errors by active learning techniques. We also compute the memory matrix to understand the memory consumption of each measurement.

Then in meta-test, we assume that we have completed meta-training, and hence know the configuration embeddings. The meta-test goal is to predict the tradeoff between memory usage and test error on a new (meta-test) dataset. This step corresponds to inference in traditional machine learning; it must be quick and cheap to satisfy the needs of resource-constrained practitioners. To select the configuration that has smallest test error and takes less than the memory limit, we measure a few selected (informative and cheap) configurations on the meta-test dataset, and use the information of their test errors to predict the rest. The problem of choosing measurements is known as the "cold-start" problem in the literature on recommender systems. Then we use the model built on the meta-training datasets to estimate values for the other measurements on the meta-test dataset.

### 3.1 META-TRAINING

On the $n$ meta-training datasets, we first theoretically compute the memory needed to evaluate each of the $d$ low-precision configurations to obtain the full memory matrix $M \in \mathbb{R}^{n \times d}$. The total memory consists of memory needed for model weights, activations, and the optimizer (gradients, gradient

accumulators, etc.) (Sohoni et al., 2019b). Among these three types, activations typically dominate, as shown in Figure 5. Thus using lower precision for activations drastically reduces memory usage. Additionally, empirical studies (Zhou et al., 2016) have shown that adopting higher precision formats for the optimizer can substantially improve the accuracy of the trained model. Since the optimizer usually requires a relatively small memory, it is often the best to use a higher precision format for this component. Thus for each low-precision configuration, it is typical to use a lower precision for network weights and activations, and a higher precision for the optimizer, resulting in combinatorially many choices in total. An example of the memory usage in this low-precision scenario is also shown in Figure 5.

In practice, ML tasks are often correlated: for example, meta-training datasets might be subsets of a large dataset like CIFAR-100 or ImageNet. The ranking of different configurations tends to be similar on similar tasks. For example, we computed the test errors of 99 low-precision configurations on 87 datasets (both listed in Appendix A) to form a performance vector in $\mathbb{R}^{99}$ for each dataset. We use the Kendall tau correlation to characterize the alignment between the ranking of errors incurred by different configurations on two datasets: the Kendall tau correlation is 1 if the order is the same and -1 if the order is reversed. As shown in Figure 6(a), similar datasets have larger correlations: for example, datasets with indices 38–87 correspond to ImageNet subproblems such as distinguishing types of fruit or boat. Notice also that some dataset pairs have configuration performance rankings that are negatively correlated. The corresponding error matrix $E$ concatenates the performance vector for each dataset. It is not low rank, but its singular values decay rapidly, as shown in Figure 6(b). Hence we expect low rank approximation of this matrix from a few measurements to work well: it is not necessary to measure every configuration.

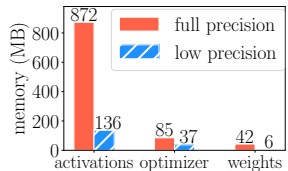

Figure 5: Memory usage under two training paradigms. Both train a ResNet-18 on CIFAR-10 with batch size 32.

In the uniform sampling scheme, we sample measurements uniformly at random to obtain a partially observed error matrix $P_\Omega(E)$. We then estimate the full error matrix $E$ using a low rank matrix completion method to form estimate $\widehat{E}$. In this paper, we use SOFTIMPUTE (Mazumder et al., 2010; Hastie et al., 2015) (Algorithm 1, Appendix B). Using the estimated error matrix $\widehat{E}$ and computed memory matrix $M$, we compute the Pareto frontier for each dataset to understand the (estimated) error-memory tradeoff. Section 4 Figure 10 shows an example.

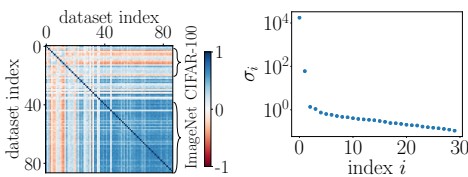

(a) dataset correlation    (b) singular value decay

In the non-uniform sampling scheme, we sample measurements with non-uniform probabilities, and use a weighted variant of SOFTIMPUTE, weighting by the inverse sampling probability, to complete the error matrix (Ma & Chen, 2019). Then we estimate

Figure 6: Kendall tau correlation of test error of all configurations between all pairs of datasets, and singular value decay of corresponding error matrix. Strong correlations allow PEPPP to succeed with a few measurements. Details in Appendix A.

the Pareto frontier in the same way as above. Non-uniform sampling is useful to reduce the memory needed for measurements. To this end, we construct a probability matrix $P$ by entry-wise transformation $P_{ij} = \sigma(1/M_{ij})$, in which the monotonically increasing $\sigma : \mathbb{R} \to [0, 1]$ maps inverse memory usage to a sampling probability. In this way, we make more low-memory measurements, and thus reduce the total memory usage.

## 3.2 META-TEST

Having estimated the embedding of each configuration, our goal is to quickly compute the error-memory tradeoff on the meta-test dataset, avoiding the exhaustive search of all possible measurements.

We first compute the memory usage of each low-precision configuration on the meta-test dataset. With this information and guided by meta-training, we evaluate only a few (cheap but informative) configurations and predict the rest. Users then finally select the non-dominated configuration with highest allowable memory. An example of the process is shown in Figure 7.

PEPPP uses Experiment Design with Matrix Factorization (ED-MF) described below to choose informative configurations. With more time, ED-MF evaluates more configurations, improving our estimates. We may also set a hard cutoff on memory: we do not evaluate low-precision configurations exceeding the memory limit. This setting has been studied in different contexts, and is called *active learning* or *sequential decision making*.

ED-MF picks measurements to minimize the variance of the resulting estimate. Specifically, we factorize the true (or estimated) error matrix $E \in \mathbb{R}^{n \times d}$ (or $\widehat{E}$) into its best rank-$k$ approximation, and get dataset embeddings $X \in \mathbb{R}^{n \times k}$ and configuration embeddings $Y \in \mathbb{R}^{d \times k}$ as $E \approx X^\top Y$ (Fusi et al., 2018; Yang et al., 2019). On a meta-test dataset, we denote the error and memory vectors of the configurations as $e^{\text{new}} \in \mathbb{R}^d$ and $m^{\text{new}} \in \mathbb{R}^d$, respectively. We model the error on the new dataset as $e^{\text{new}} = Y^\top x^{\text{new}} + \epsilon \in \mathbb{R}^d$, where $x^{\text{new}} \in \mathbb{R}^k$ is the embedding of the new dataset and $\epsilon \in \mathbb{R}^d$ accounts for the errors from both measurement and low rank decomposition. We estimate the embedding $x^{\text{new}}$ from a few measurements (entries) of $e^{\text{new}}$ by least squares. Note that this requires at least $k$ measurements on the meta-test dataset to make meaningful estimations, in which $k$ is the rank for matrix factorization.

If $\epsilon \sim \mathcal{N}(0, \sigma^2 I)$, the variance of the estimator is $\left( \sum_{j \in S} y_j y_j^\top \right)^{-1}$, in which $y_j$ is the $j$th column of $Y$. D-optimal experiment design selects measurements on the new dataset by minimizing (a scalarization of) the variance,

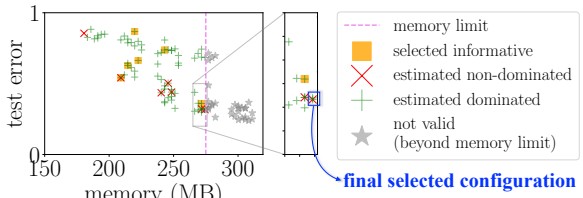

$$\begin{aligned} \text{minimize} \quad & \log \det \left( \sum_{j \in S} y_j y_j^\top \right)^{-1} \\ \text{subject to} \quad & |S| \leq l \\ & S \subseteq [d], \end{aligned}$$

$$(1)$$

to find $S$, the set of indices of configurations to evaluate. The positive integer $l$ bounds the number of measurements. Given a memory cap $m_{\text{max}}$, we replace the second constraint above by $S \subseteq T$, in which $T = \{j \in [d] \mid m_j^{\text{new}} \leq m_{\text{max}}\}$ is the set of feasible configurations. Since Problem 1 is combinatorially hard, we may either relax it to a convex optimization problem by allowing decision variables to have non-integer values between 0 and 1, or use a greedy method to incrementally choose measurements (initialized by measurements chosen by the column-pivoted QR decomposition (Gu & Eisenstat, 1996; Golub & Van Loan, 2012)). We compare these two approaches in Appendix C.

Figure 7: Meta-test on CIFAR-10. After meta-training on all other datasets in Appendix A Table 2, we use ED-MF to choose six informative measurements (orange squares) with a 275MB memory limit for each measurement on CIFAR-10. Then we estimate test errors of other configurations by ED-MF, and restrict our attention to configurations that we estimate to be non-dominated (red x's). Note some of these are in fact dominated, since we plot true (not estimated) test error! Finally we select the estimated non-dominated configuration with highest allowable memory (blue square).

In Section 4.2, we compare ED-MF with competing techniques and show it works the best to estimate the Pareto frontier and select the final configuration.

## 4 EXPERIMENTS AND DISCUSSIONS

The code for PEPPP and experiments is in the GitHub repository at `https://github.com/chengrunyang/peppp`. We use QPyTorch (Zhang et al., 2019) to simulate low-precision formats on standard hardware. The 87 datasets and 99 low-precision configurations in experiments are listed in Appendix A. The datasets consist of natural and medical images from various domains. Apart from CIFAR-10, the datasets include 20 CIFAR-100 partitions from mutually exclusive subsets, based on the superclass labels. They also include 50 subsets of ImageNet (Deng et al., 2009), which contains over 20,000 classes grouped into multiple major hierarchical categories like fungus and amphibian; each of the 50 datasets come from different hierarchies. Finally, we use 16 datasets from the visual domain benchmark (Wallace & Hariharan, 2020) to increase the diversity of domains.

The 99 low-precision configurations use mixed-precision as described in Section 3.1. Format A (for the activations and weights) uses 5 to 9 bits; Format B (for the optimizer) ranges from 14 to

20 bits. For each, these bits may be split arbitrarily between the exponent and mantissa. Across all configurations, we use ResNet-18, ResNet-34 and VGG (Simonyan & Zisserman, 2014) variants (11, 13, 16, 19) with learning rate 0.001, momentum 0.9, weight decay 0.0005 and batch size 32. Each training uses only 10 epochs as an early stopping strategy (Yao et al., 2007) to prevent overfitting. All the measurements take 35 GPU days on NVIDIA® GeForce® RTX 3090.

Developing a method to select the above hyperparameters at the same time as the low-precision format is an important topic for future research, but not our focus here. In Section 4.3 we demonstrate how PEPPP can naturally extend to efficiently select both optimization and low-precision hyperparameters.

A number of metrics may be used to evaluate the quality of the obtained Pareto frontier in a multi-objective optimization problem (Wu & Azarm, 2001; Li et al., 2014; Audet et al., 2020): the distance between approximated and true frontiers (convergence), the uniformity of the distances between neighboring Pareto optimal points (uniformity), how well the frontier is covered by the approximated points (spread), etc. In our setting, the memory is accurate and the test error is estimated, so we do not have full control of the uniformity and spread of the 2-dimensional frontiers between test error and memory. As illustrated in Figure 8, we evaluate the quality of our estimated Pareto frontiers by the following two metrics:

**Convergence** (Deb et al., 2002) between the sets of true and estimated Pareto optimal points $\mathcal{P}$, $\widehat{\mathcal{P}}$ is $\frac{1}{|\widehat{\mathcal{P}}|} \sum_{v \in \widehat{\mathcal{P}}} \mathrm{dist}(v, \mathcal{P})$, where the distance between point $v$ and set $\mathcal{P}$ is $\mathrm{dist}(v, \mathcal{P}) = \min\{\|v - w\| : w \in \mathcal{P}\}$. This is a surrogate for the distance between Pareto frontiers.

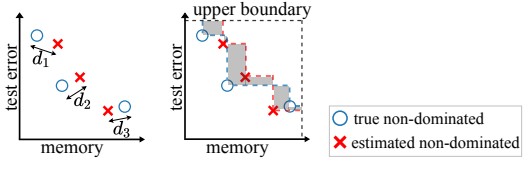

(a) convergence  (b) HyperDiff

**Hypervolume difference (HyperDiff)** (Zitzler & Thiele, 1998; Coello & Sierra, 1999; Wu & Azarm, 2001) is the absolute difference between the volumes of solution spaces dominated by the true and estimated Pareto frontiers. This metric improves with better convergence, uniformity, and spread. Its computation requires an upper boundary for each resource.

Figure 8: Illustration of Pareto frontier metrics. (a) Convergence is the average distance from each estimated Pareto optimal point to its closest true point: $\mathrm{average}(d_1, d_2, d_3)$. (b) HyperDiff is the absolute difference in area of feasible regions given by the true and estimated Pareto optimal points: the shaded area between Pareto frontiers.

When computing these metrics, we normalize the memories by proportionally scaling them to between 0 and 1. To evaluate the matrix completion performance, we use the relative error defined as $\|\widehat{v} - v\|/\|v\|$, in which $\widehat{v}$ is the predicted vector and $v$ is the true vector.

## 4.1 META-TRAINING

We study the effectiveness of uniform and non-uniform sampling schemes. For simplicity, we regard all 87 available datasets as meta-training in this section. In SOFTIMPUTE, we use $\lambda = 0.1$ (chosen by cross-validation from a logarithmic scale); and we choose a rank 5 approximation, which accounts for 78% of the variance in the error matrix (as shown in Figure 6(b)).

In the uniform sampling scheme, we investigate how many samples we need for accurate estimates: we sample the error matrix at a number of different ratios, ranging from 5% to 50%, and complete the error matrix from each set of sampled measurements. In uniform sampling, the sampling ratio is also the percentage of memory we need to make the measurements in parallel, compared to exhaustive search. Hence we show the relationship between Pareto frontier estimation performance and sampling ratio in Figure 9. We can see the estimates are more accurate at larger sampling ratios, but sampling 20% of the entries already suffices for good performance.

As a more intuitive example, Figure 10 shows the error-memory tradeoff on CIFAR-10. Compared to the estimated Pareto optimal points at sampling ratio 5%, the ones at 20% are closer to the true Pareto frontier (better convergence), lie more evenly (better uniformity) and cover the true Pareto frontier better (better spread), as shown by the respective convergence and HyperDiff values.

The non-uniform sampling scheme has a similar trend and is shown in Appendix E.2.

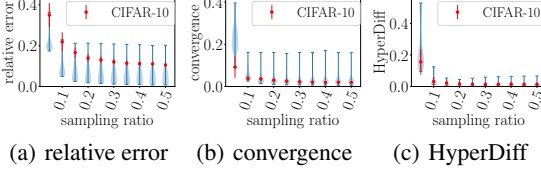

(a) relative error    (b) convergence    (c) HyperDiff

Figure 9: Pareto frontier estimation in PEPPP meta-training, with uniform sampling of configurations. The violins show the distribution of the performance on individual datasets, and the error bars (blue) show the range. The red error bars show the standard deviation of the error on CIFAR-10 across 100 random samples of the error matrix. Figure (a) shows the matrix completion error for each dataset; Figure (b) and (c) show the performance of the Pareto frontier estimates. Modest sampling ratios (around 0.1) already yield good performance.

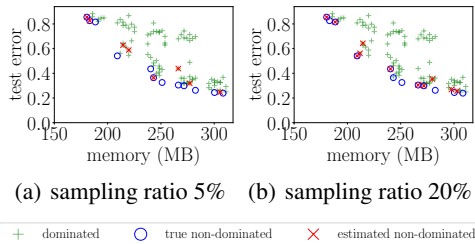

(a) sampling ratio 5%    (b) sampling ratio 20%

+ dominated    ○ true non-dominated    ✕ estimated non-dominated

Figure 10: Error vs memory on CIFAR-10 with true and estimated Pareto frontiers from uniform sampling in PEPPP meta-training. A 20% uniform sample of entries yields a better estimate of the Pareto frontier (convergence 0.03 and HyperDiff 0.02) compared to a 5% sample (convergence 0.09 and HyperDiff 0.16).

## 4.2 META-LEAVE-ONE-OUT CROSS-VALIDATION (META-LOOCV)

Now suppose that we have already collected measurements on the meta-training datasets to form a meta-training error matrix $E$ (or its low rank approximation $\widehat{E}$). On the meta-test dataset, PEPPP estimates the Pareto frontier by the active learning technique ED-MF. We compare ED-MF with a few other active learning techniques: Random selection with matrix factorization (RANDOM-MF), QR decomposition with column pivoting and matrix factorization (QR-MF) and two Bayesian optimization techniques (BO-MF and BO-FULL), to understand whether the strong assumptions in ED-MF (low rank and Gaussian errors) are a hindrance or a help. An introduction to these techniques can be found in Appendix E.1.

We use rank 3 for matrix factorization: $Y_{:,j} \in \mathbb{R}^3$ for each $j \in [d]$. In BO, we tune hyperparameters on a logarithmic scale and choose the RBF kernel with length scale 20, white noise with variance 1, and $\xi = 0.01$. Table 1 shows the meta-LOOCV settings for each acquisition technique. We compare the techniques at a range of number of configurations to measure in each meta-LOOCV split,

Table 1: Meta-LOOCV experiment settings

| meta-training error matrix | memory cap on meta-test? | |
|---|---|---|
| | no | yes |
| full | **I** | II |
| uniformly sampled | III | **IV** |
| non-uniformly sampled | V | VI |

resembling what practitioners do in hyperparameter tuning: evaluate an informative subset and infer the rest. Setting I is the most basic, Setting IV and VI are the most practical. We only show results of Setting I and IV in the main paper, and defer the rest to Appendix E.

In Setting I, we do meta-LOOCV with the full meta-training error matrix in each split and do not cap the memory for meta-test. This means we evaluate every configuration on the meta-training datasets. We can see from Figure 11(a) and 11(b) that:

- ED-MF stably outperforms under both metrics, especially with fewer measurements.

- QR-MF overall matches the performance of ED-MF.

- BO-MF, BO-FULL and RANDOM-MF underperform ED-MF and QR-MF at lower memory usage, but often match their performance with higher memory.

Practitioners may have only collected part of the meta-training performance, and desire or are limited by a memory cap when they do meta-test on the new dataset. In Setting IV, we cap the single-configuration memory usage for meta-test at 816MB, the median memory of all possible measurements across configurations and datasets. Additionally, we uniformly sample 20% configurations from the meta-training error matrix in each split. In Figure 11(c) and 11(d), we can see similar trends as Setting I, except that QR-MF is slightly worse.

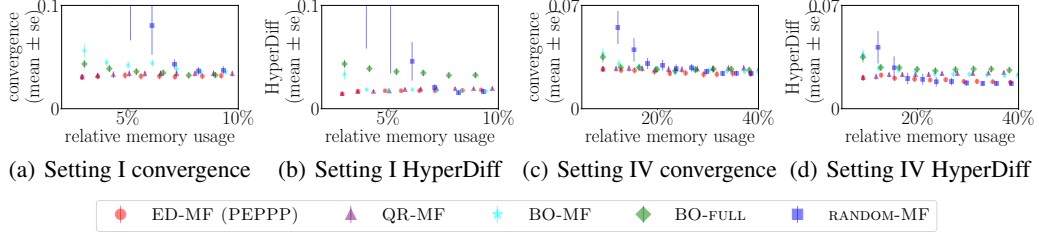

(a) Setting I convergence  (b) Setting I HyperDiff  (c) Setting IV convergence  (d) Setting IV HyperDiff

● ED-MF (PEPPP)   ▲ QR-MF   ✳ BO-MF   ◆ BO-FULL   ■ RANDOM-MF

Figure 11: Pareto frontier estimates in meta-LOOCV Setting I and IV (with a 20% meta-training sampling ratio and an 816MB meta-test memory cap). Each error bar is the standard error across datasets. The x axis measures the memory usage relative to exhaustively searching the permissible configurations. ED-MF consistently picks the configurations that give the best PF estimates.

Ultimately, users would want to select a configuration that both achieves a small error and takes lower memory than the limit. As shown in Figure 2, PEPPP offers users the predicted Pareto frontier and chooses the non-dominated configuration with the highest allowable memory and hence the lowest error. We compare the acquisition techniques with the "random high-memory" baseline that randomly chooses a configuration that takes the highest allowable memory: an approach that follows the "higher memory, lower error" intuition. Figure 12 shows an example.

Overall, among all approaches and across all memory usage, ED-MF outperforms, especially at a smaller number of measurements. Compared to BO techniques, ED-MF also enjoys the benefit of having less hyperparameters to tune. Although techniques like QR-MF and RANDOM-MF are easier to implement, the additional cost of ED-MF is much smaller than the cost of making measurements: neural network training and testing.

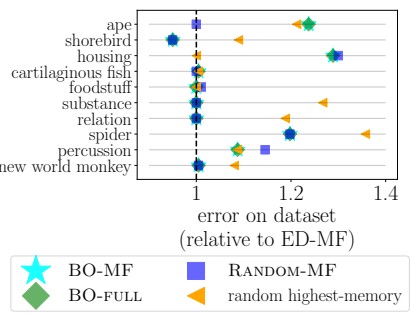

BO-MF   RANDOM-MF
BO-FULL   random highest-memory

Figure 12: Relative performance with respect to ED-MF in meta-test Setting IV when making 3 measurements (memory usage ∼10%) on 10 ImageNet partitions. ED-MF outperforms in most cases.

### 4.3 TUNING OPTIMIZATION HYPERPARAMETERS

Previously, we have shown that PEPPP can estimate the error-memory tradeoff and select a promising configuration with other hyperparameters fixed. In practice, users may also want to tune hyperparameters like learning rate to achieve the lowest error. In Appendix E.4, we tune the number of epochs and learning rate in addition to precision, and show that the methodology can be used in broader settings.

### 4.4 META-LEARNING ACROSS ARCHITECTURES

The meta-learning in previous sections were conducted on ResNet-18. In Appendix E.5, we show that on 10 ImageNet partitions, PEPPP is also capable of estimating the error-memory tradeoff of ResNet-34, VGG-11, VGG-13, VGG-16 and VGG-19 competitively. Moreover, the meta-learning across architectures works better than considering each architecture separately.

## 5 CONCLUSION

This paper proposes PEPPP, a meta-learning system to select low-precision configurations that leverages training information from related tasks to efficiently pick the perfect precision given a memory budget. Built on low rank matrix completion with active learning, PEPPP estimates the Pareto frontier between memory usage and model performance to find the best low-precision configuration at each memory level. By reducing the cost of hyperparameter tuning in low-precision training, PEPPP allows practitioners to efficiently train accurate models.

ACKNOWLEDGMENTS

Madeleine Udell and Chengrun Yang acknowledge support from NSF Award IIS-1943131, the ONR Young Investigator Program, and the Alfred P. Sloan Foundation. Christopher De Sa acknowledges a gift from SambaNova Systems, Inc., and Jerry Chee acknowledges support from the NSF Research Traineeship Program (Award # 1922551). The authors thank Tianyi Zhang for helpful discussions, and thank several anonymous reviewers for useful comments.

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

All the tables in this appendix are shown in the last page.

## A    DATASETS AND LOW-PRECISION FORMATS

The 87 datasets chosen in our experiment are listed in Table 2. Note that the datasets contain images of two different resolutions: 32 denotes resolution $3 \times 32 \times 32$, and 64 denotes $3 \times 64 \times 64$. We list our choices of Format A and B in Table 3 and 4, respectively. In each evaluation, the low-precision configuration is composed of one Format A (for the activations and weights) and one Format B (for the optimizer), as detailed in Section 3.1. Therefore, the total number of low-precision configurations used in our experiment is 99.

## B    THE SOFTIMPUTE ALGORITHM

There are several versions of SOFTIMPUTE; (Hastie et al., 2015) gives a nice overview. We use the version in (Mazumder et al., 2010).

At a high level, SOFTIMPUTE iteratively applies a soft-thresholding operator $\mathcal{S}_\lambda$ on the partially observed error matrix with a series of decreasing $\lambda$ values. Each $\mathcal{S}_\lambda$ replaces the singular values $\{\sigma_i\}$ with $\{(\sigma_i - \lambda)_+\}$. The regularization parameter $\lambda$ can be set in advance or ad hoc, by convergence dynamics.

The pseudocode for the general SOFTIMPUTE algorithm is shown as Algorithm 1, in which $P_\Omega(E)$ is a matrix with the same shape as $E$, and has the $(i,j)$-th entry being $E_{ij}$ if $(i,j) \in \Omega$, and 0 otherwise. In our implementation, $\lambda_i = \lambda t_i$ for each step $i$, and $t_i$ is the step size from TFOCS backtracking (Becker et al., 2011).

---
**Algorithm 1** SOFTIMPUTE

**Input:** a partially observed matrix $P_\Omega(E) \in \mathbb{R}^{n \times d}$, number of iterations $I$, a series of decreasing $\lambda$ values $\{\lambda_i\}_{i=1}^I$
**Output:** an estimate $\widehat{E}$
1  **for** $i = 1$ **to** $I$ **do**
2  $\quad \widetilde{E} \leftarrow P_\Omega(E) + P_{\Omega^C}(\widehat{E})$
3  $\quad U, \Sigma, V \leftarrow \mathrm{svd}(\widetilde{E})$
4  $\quad \widehat{E} \leftarrow U\mathcal{S}_\lambda(\Sigma)V^\top$

---

## C    ALGORITHMS FOR EXPERIMENT DESIGN

As mentioned in Section 3.2, there are mainly two algorithms to solve Problem 1, the D-optimal experiment design problem: convexification and greedy.

The convexification approach relaxes the combinatorial optimization problem to the convex optimization problem

$$\begin{aligned} \text{minimize} \quad & \log\det\left(\sum_{j=1}^d v_j y_j y_j^\top\right)^{-1} \\ \text{subject to} \quad & \sum_{j=1}^d v_j \leq l \\ & v_j \in [0,1], \forall j \in [d] \end{aligned} \tag{2}$$

that can be solved by a convex solver (like SLSQP). Then we sort the entries in the optimal solution $v^* \in \mathbb{R}^d$ and set the largest $l$ entries to 1 and the rest to 0.

The greedy approach (Madan et al., 2019; Yang et al., 2020) maximizes the submodular objective function by first choosing an initial set of configurations by column-pivoted QR decomposition, and then greedily adding new configuration to the solution set $S$ in each step until $|S| = l$. The greedy stepwise selection algorithm is shown as Algorithm 2, in which the new configuration is chosen by

the Matrix Determinant Lemma[*] (Harville, 1998), and $X_t^{-1}$ is updated by the close form from the Sherman-Morrison Formula[*] (Sherman & Morrison, 1950). The column-pivoted QR decomposition selects top $k$ pivot columns of $Y \in \mathbb{R}^{k \times d}$ to get the index set $S_0$, ensuring that $X_0 = \sum_{j \in S_0} y_j y_j^\top$ is non-singular.

---

**Algorithm 2** Greedy algorithm for D-optimal experiment design

---

**Input:** design vectors $\{y_j\}_{j=1}^d$, in which $y_j \in \mathbb{R}^k$; maximum number of selected configurations $l$;
    initial set of configurations $S_0 \subseteq [d]$, s.t. $X_0 = \sum_{j \in S_0} y_j y_j^\top$ is non-singular
**Output:** the selected set of designs $S \subseteq [d]$
1   $S \leftarrow S_0$
2   **while** $|S| \leq l$ **do**
3      $i \leftarrow \mathrm{argmax}_{j \in [d] \backslash S} y_j^\top X_t^{-1} y_j$
4      $S \leftarrow S \cup \{i\}$
5      $X_{t+1} \leftarrow X_t + y_i y_i^\top$

---

The convexification approach empirically works because most entries of the optimal solution $v^*$ are close to either 0 or 1. The histogram in Figure 13 shows an example when we use rank $k = 5$ to factorize the entire error matrix and set $l = 20$.

In terms of solution quality, the relative error plot in Figure 13 shows that the greedy approach consistently outperforms in terms of the relative matrix completion error for each dataset in ED-MF solutions. Since the greedy approach is also more than $10\times$ faster than convexification (implemented by `scipy`), we use the greedy approach throughout all experiments for the rest of this paper.

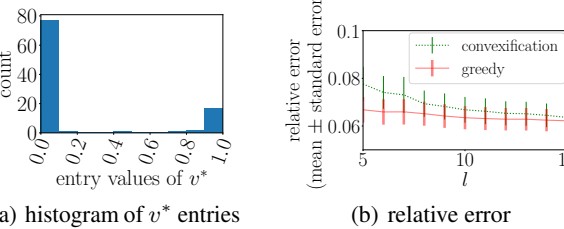

(a) histogram of $v^*$ entries       (b) relative error

Figure 13: Convexification vs greedy for ED.

## D  INFORMATION TO HARDWARE DESIGN

Among all 87 datasets (each has its own error-memory tradeoff), the promising configurations that we identify among all 99 configurations are as below (number of sign bits - number of exponent bits -number of mantissa bits).

- Format A: 1-4-1, Format B: 1-6-7 (appears on 60 out of 87 Pareto frontiers)
- Format A: 1-4-1, Format B: 1-7-7 (appears on 33 out of 87 Pareto frontiers)
- Format A: 1-3-1, Format B: 1-7-7 (appears on 30 out of 87 Pareto frontiers)
- Format A: 1-4-2, Format B: 1-6-7 (appears on 24 out of 87 Pareto frontiers)
- Format A: 1-5-2, Format B: 1-6-7 (appears on 21 out of 87 Pareto frontiers)

## E  MORE DETAILS ON EXPERIMENTS

We first show the plot of explained variances of top principal components in Figure 14: how much variance in our data do the first several principal components account for (Bishop, 2006). This quantity is computed by the ratio of sum of squares of the first $k$ singular values to that of all singular values. In Figure 14, we vary $k$ from 1 to 30, corresponding to the decay of singular values shown in Figure 6(b). We can see the first singular value already accounts for $99.0\%$ of the total variance, and the first two singular values account for more than $99.5\%$. This means we can pick a small rank for PCA in meta-training and still keep the most information in our meta-training data.

---

[*]The Matrix Determinant Lemma states that for any invertible matrix $A \in \mathbb{R}^{k \times k}$ and $a, b \in \mathbb{R}^k$, $\det(A + ab^\top) = \det(A)(1 + b^\top A^{-1} a)$. Thus $\mathrm{argmax}_{j \in [d] \backslash S} \det(X_t + y_j y_j^\top) = \mathrm{argmax}_{j \in [d] \backslash S} y_j^\top X_t^{-1} y_j$.

[*]The Sherman-Morrison Formula states that for any invertible matrix $A \in \mathbb{R}^{k \times k}$ and $a, b \in \mathbb{R}^k$, $(A + ab^\top)^{-1} = A^{-1} - \frac{A^{-1} ab^\top A^{-1}}{1 + b^\top A^{-1} a}$.

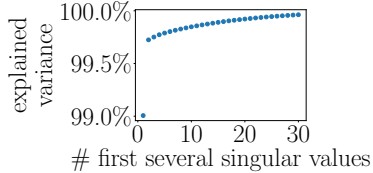

Figure 14: Explained variance of the first several singular values in Figure 6(b).

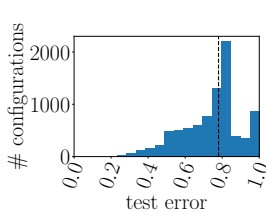
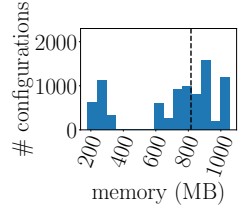
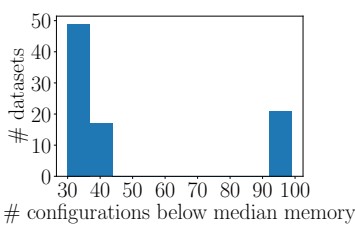

Figure 15: Histograms of error and memory. The dashed lines are the respective medians.

Figure 16: Histogram of datasets by the number of configurations that take memories less than the overall median of 816MB.

We use the ratio of incorrectly classified images as our error metric. Figure 15 shows histograms of error and memory values in our error and memory matrices from evaluating 99 configurations on 87 datasets. The vertical dashed lines show the respective medians: 0.78 for test error and 816MB for memory. We can see both the test error and memory values span a wide range. The errors come from training a wide range of low-precision configurations with optimization hyperparameters not fine-tuned, and are thus larger than SOTA results. In Section 4.3 of the main paper and Section E.4 here, we show that training for a larger number of epochs and with some other optimization hyperparameters yield the same error-memory tradeoff as Figure 1(a) in the main paper.

In meta-LOOCV settings with a meta-test memory cap at the median memory 816MB, Figure 16 shows a histogram of number of feasible configurations on each of the 87 datasets. There are "cheap" (resolution 32) datasets on which each of the 99 configurations takes less than the cap, and "expensive" (resolution 64) datasets on which the feasible configurations are far less than 99.

### E.1 INTRODUCTION TO RANDOM-MF, QR-MF AND BO

- **Random selection with matrix factorization (RANDOM-MF).** Same as ED-MF, RANDOM-MF predicts the unevaluated configurations by linear regression, except that it selects the configurations to evaluate by random sampling.
- **QR decomposition with column pivoting and matrix factorization (QR-MF).** QR-MF first selects the configurations to evaluate by QR decomposition with column pivoting: $EP = QR$, in which the permutation matrix $P$ gives the most informative configurations. Then it predicts unevaluated configurations in the same way as ED-MF and RANDOM-MF.
- **Bayesian optimization (BO).** Bayesian optimization is a sequential decision making framework that learns and optimizes a black-box function by incrementally building surrogate models and choosing new measurements (Frazier, 2018). It works best for black-box functions that are expensive to evaluate and lack special structures or derivatives. We compare ED-MF with two BO techniques. The first technique, BO-MF, applies BO to the function $f_1 : \mathbb{R}^k \to \mathbb{R}$ that maps low-dimensional configuration embeddings $\{Y_{:,j}\}_{j=1}^d$ to the test errors of the configurations on the meta-test dataset $\{e_j^{\text{new}}\}_{j=1}^d$ (Fusi et al., 2018). The embeddings come from the same low-rank factorization of the error matrix $E$ as in ED-MF. The second, BO-FULL, applies BO to the function $f_2 : \mathbb{R}^n \to \mathbb{R}$ that directly maps columns of the error matrix $\{E_{:,j}\}_{j=1}^d$ to $\{e_j^{\text{new}}\}_{j=1}^d$. To learn either of these black-box functions, we start by evaluating a subset of configurations $S \subseteq [d]$ and then incrementally select new configurations that maximize the expected improvement (Močkus, 1975; Jones et al., 1998).

### E.2 ADDITIONAL META-TRAINING RESULTS: NON-UNIFORM SAMPLING

In non-uniform sampling, we sample each entry of the error matrix $E$ with probabilities $P_{ij} = \sigma(1/W_{ij})$, in which $\sigma$ maps $\{1/W_{ij}\}$ into an interval $[0, p_{\max}] \subseteq [0, 1]$ according to the cumulative distribution function of $\{1/W_{ij}\}$. By varying $p_{\max}$, we change how we aggressively sample the configurations, how different the sampling probabilities are between large and small memory configu-

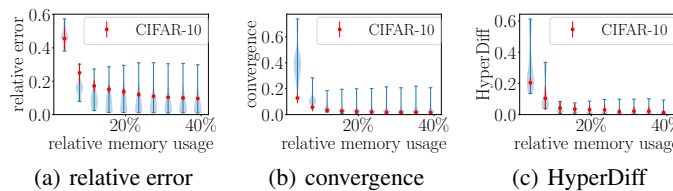

(a) relative error     (b) convergence     (c) HyperDiff

Figure 17: Pareto frontier estimation performance in PEPPP meta-training with non-uniform sampling of configurations. The violins and scatters have the same meaning as Figure 9. The x axis measures the memory usage relative to exhaustive search.

rations, and also the percentage of memory needed. In Figure 17, we vary $p_{\max}$ from 0.1 to 1 and see that the quality of the estimated Pareto frontier improves with more memory.

### E.3 ADDITIONAL META-LOOCV RESULTS

In the main paper, we have shown the performance of Pareto frontier estimates and configuration selection for Setting I and IV. For the rest of the settings in Table 1, we conduct meta-LOOCV in the same way and show the results of Pareto frontier estimates in Figure 18.

**Setting II.** We have the 816MB memory cap for meta-test, and have the full meta-training error matrix in each split.

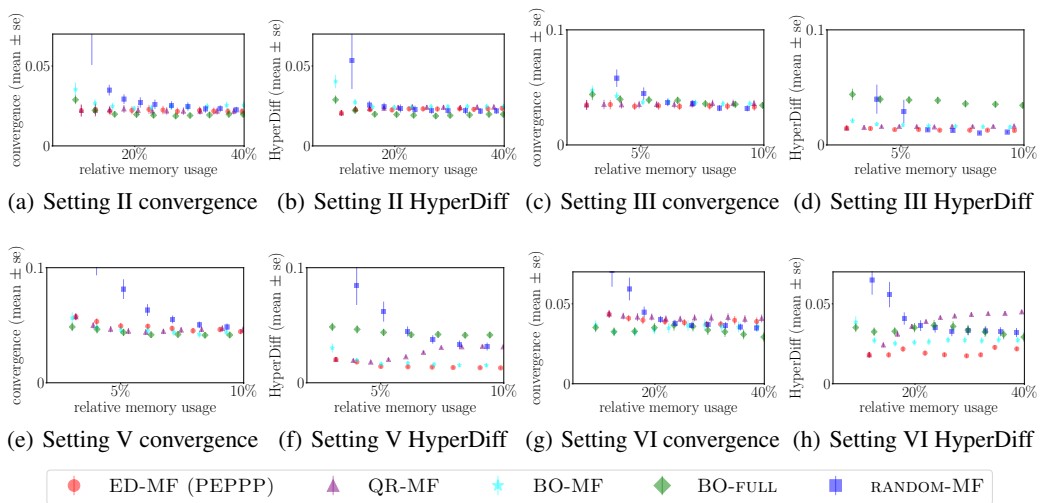

(a) Setting II convergence   (b) Setting II HyperDiff   (c) Setting III convergence   (d) Setting III HyperDiff

(e) Setting V convergence   (f) Setting V HyperDiff   (g) Setting VI convergence   (h) Setting VI HyperDiff

● ED-MF (PEPPP)   ▲ QR-MF   ★ BO-MF   ◆ BO-FULL   ■ RANDOM-MF

Figure 18: Pareto frontier estimates in meta-LOOCV Setting II (full meta-training error matrix, a 816MB memory cap), Setting III (uniformly sample 20% meta-training measurements, no meta-test memory cap), Setting V (non-uniformly sample 20% meta-training measurements, no meta-test memory cap), and Setting VI (non-uniformly sample 20% meta-training measurements, an 816MB meta-test memory cap). Each error bar is the standard error across datasets. ED-MF is among the best in every setting and under both metrics.

**Setting III.** We have no memory cap for meta-test, and uniformly sample 20% configurations for meta-training in each split.

**Setting V.** We have no memory cap for meta-test, and non-uniformly sample 20% configurations for meta-training in each split. The sampling method is the same as in meta-training (Figure 17).

**Setting VI.** We have the 816MB memory cap for meta-test, and non-uniformly sample 20% meta-training configurations in the same way as Setting V above.

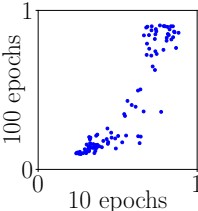

Figure 19: Errors of 99 configurations trained for different numbers of epochs.

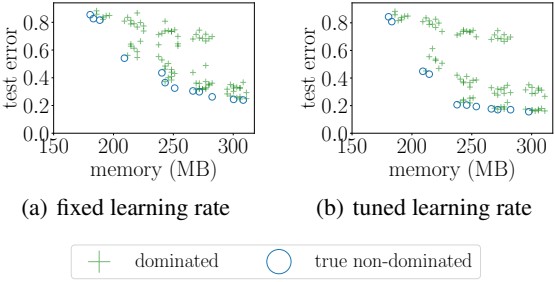

(a) fixed learning rate    (b) tuned learning rate

+ dominated    ○ true non-dominated

Figure 20: CIFAR-10 error-memory tradeoff. Figure (a) has learning rate 0.001 for all low-precision configurations. Figure (b) shows the tradeoff with tuned learning rates: at each low-precision configuration, the lowest test error achieved by learning rates $\{0.01, 0.001, 0.0001\}$ is selected.

## E.4    TUNING OPTIMIZATION HYPERPARAMETERS

### E.4.1    NUMBER OF EPOCHS

The low-precision networks still underfit after 10 epochs of training. This situation is typical: underfitting due to budget constraints is unfortunately common in deep learning. Luckily, meta-learning the best precision will succeed so long as the validation errors are correctly *ordered*, even if they all overestimate the error of the corresponding fully trained model. Indeed, our validation errors correlate well with the errors achieved after further training: on CIFAR-10, the Kendall tau correlation between ResNet-18 errors at 10 epochs and 100 epochs is 0.73, shown in Figure 19. The lowest error at 100 epochs is 9.7%, only approximately 2% higher than SOTA (Bungert et al., 2021).

### E.4.2    LEARNING RATE

Previously, we have shown that PEPPP can estimate the error-memory tradeoff and select a promising configuration with other hyperparameters fixed. In practice, users may also want to tune hyperparameters like learning rate to achieve the lowest error. Here, we tune learning rate in addition to precision, and show that the methodology can be used in broader settings of hyperparameter tuning.

Figure 20 shows that the error-memory tradeoff still exists with a fine-tuned learning rate for each configuration. With the learning rate tuned across $\{0.01, 0.001, 0.0001\}$, the test errors are in general smaller, but high-memory configurations still achieve lower errors in general. Thus the need to efficiently select a low-precision configuration persists.

Our approach can naturally extend to efficiently selecting optimization *and* low-precision hyperparameters. We perform the meta-training and meta-LOOCV experiments on a subset of CIFAR-100 partitions with multiple learning rates $\{0.01, 0.001, 0.0001\}$. The error and memory matrices we use here have 45 rows and 99 columns, respectively. Learning rate and low-precision configuration are collapsed into the same dimension: each row corresponds to a combination of one CIFAR-100 subset and one of the learning rates $\{0.01, 0.001, 0.0001\}$, as shown in Table 5. We say that these error and memory matrices are *LR-tuned*. Figure 21 shows the LR-tuned error matrix also has a fast singular value decay. The other hyperparameters are the same as in LR-fixed experiments, except that we use

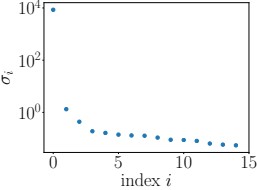

Figure 21: Singular value decay of the LR-tuned error matrix.

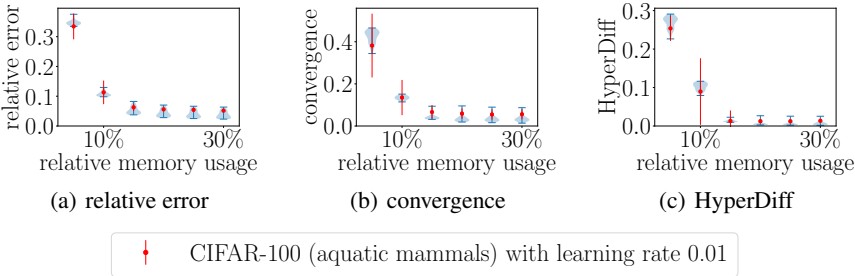

(a) relative error   (b) convergence   (c) HyperDiff

CIFAR-100 (aquatic mammals) with learning rate 0.01

Figure 22: The Pareto frontier estimation performance in meta-training, with uniform sampling of configurations on the LR-tuned error and memory matrices. Similar to Figure 9, the violins show the distribution of the performance on individual datasets, and the error bars (blue) show the range. The red error bars show the standard deviation of the error on CIFAR-100 aquatic mammals and learning rate 0.01, across 100 random samples of the error matrix. Figure (a) shows the matrix completion error for each dataset; Figure (b) and (c) show the performance of the Pareto frontier estimates in convergence and HyperDiff.

batch size 128 and train for 100 epochs. The meta-training and meta-LOOCV results are consistent with those in Sections 4.1 and 4.2, respectively:

- In meta-training, we first uniformly sample the error matrix and study the performance of matrix completion and Pareto frontier estimation. Figure 22 shows the matrix completion error and Pareto frontier estimation metrics. Then we do non-uniformly sampling and get Figure 23, the LR-tuned version of Figure 17 in the main paper.

- In meta-test, we evaluate settings in Table 1 in the main paper. We get the performance of Pareto frontier estimates in Figure 24, the LR-tuned version of Figure 11 in the main paper. ED-MF steadily outperforms.

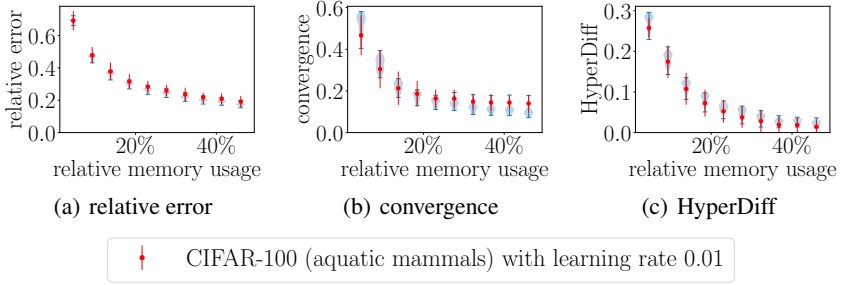

(a) relative error   (b) convergence   (c) HyperDiff

CIFAR-100 (aquatic mammals) with learning rate 0.01

Figure 23: The Pareto frontier estimation performance in meta-training, with non-uniform sampling of configurations on the LR-tuned error and memory matrices. The violins and scatters have the same meaning as Figure 17 in the main paper.

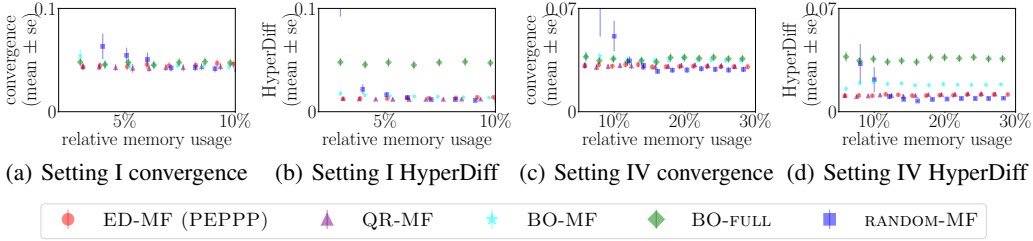

Figure 24: Pareto frontier estimates in meta-LOOCV settings on the LR-tuned error and memory matrices. Each error bar is the standard error across datasets.

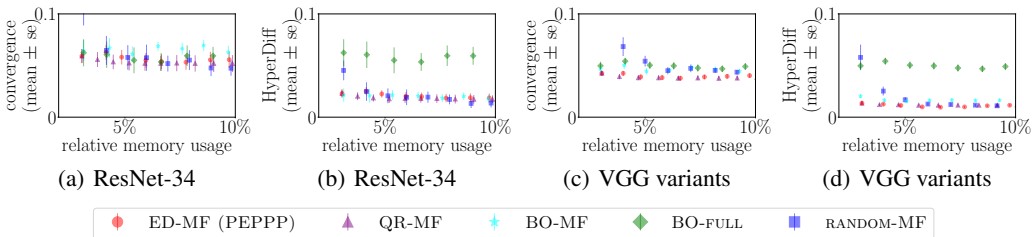

Figure 25: Pareto frontier estimates in meta-LOOCV Setting I when learning across architectures: from ResNet-18 to either ResNet-34, or to VGG variants. Each error bar is the standard error across datasets. The x axis measures the memory usage relative to exhaustively searching the permissible configurations. ED-MF consistently picks the configurations that give the best PF estimates.

### E.5 LEARNING ACROSS ARCHITECTURES

We show that on 10 ImageNet partitions, PEPPP with ED-MF is able to estimate the error-memory tradeoff of the low-precision configurations on of ResNet-34, VGG-11, VGG-13, VGG-16 and VGG-19. The 10 ImageNet partitions have WordNet IDs {n02470899, n01482071, n02022684, n03546340, n07566340, n00019613, n01772222, n03915437, n02489589, n02127808} and are randomly selected from the 50 ImageNet subsets on which we collected the error matrix. On these ImageNet partitions, we use the performance of ResNet-18 as meta-training data, and either the performance of ResNet-18 or VGG variants as meta-test data. In Figure 25, we can see that ED-MF is steadily among the best in Pareto frontier estimation, and there is no statistical difference between the estimation performance on ResNet-34 and VGG variants.

Next, we compare the performance of the following two cases:

    i Meta-learning across datasets with performance from the same architecture: For example, to learn the error-memory tradeoff of ResNet-18 on n02470899, we only use the tradeoffs of ResNet-18 on 9 other ImageNet partitions as the meta-training data.

    ii Meta-learning across datasets with performance from both the same and other architectures: For example, the error-memory tradeoff of ResNet-18 on n02470899, we not only use the tradeoffs of ResNet-18 on 9 other ImageNet partitions, but also use those of ResNet-34, VGG-11, VGG-13, VGG-16 and VGG-19 on the 9 partitions as the meta-training data.

Figure 26 shows that Case ii outperforms Case i in better estimating the error-memory tradeoffs on different architectures and datasets.

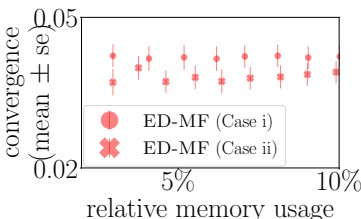

Figure 26: Benefit of meta-learning across architectures. Each error bar is the standard error across architecture-dataset combinations (e.g., ResNet-18 + n02470899 is a combination). The x axis measures the memory usage relative to exhaustively searching the permissible configurations.

Table 2: Datasets

| index | dataset name | resolution | # points |
|---|---|---|---|
| 1 | CIFAR10 | 32 | 60000 |
| 2 | CIFAR100 (aquatic mammals) | 32 | 3000 |
| 3 | CIFAR100 (fish) | 32 | 3000 |
| 4 | CIFAR100 (flowers) | 32 | 3000 |
| 5 | CIFAR100 (food containers) | 32 | 3000 |
| 6 | CIFAR100 (fruit and vegetables) | 32 | 3000 |
| 7 | CIFAR100 (household electrical devices) | 32 | 3000 |
| 8 | CIFAR100 (household furniture) | 32 | 3000 |
| 9 | CIFAR100 (insects) | 32 | 3000 |
| 10 | CIFAR100 (large carnivores) | 32 | 3000 |
| 11 | CIFAR100 (large man-made outdoor things) | 32 | 3000 |
| 12 | CIFAR100 (large natural outdoor scenes) | 32 | 3000 |
| 13 | CIFAR100 (large omnivores and herbivores) | 32 | 3000 |
| 14 | CIFAR100 (medium-sized mammals) | 32 | 3000 |
| 15 | CIFAR100 (non-insect invertebrates) | 32 | 3000 |
| 16 | CIFAR100 (people) | 32 | 3000 |
| 17 | CIFAR100 (reptiles) | 32 | 3000 |
| 18 | CIFAR100 (small mammals) | 32 | 3000 |
| 19 | CIFAR100 (trees) | 32 | 3000 |
| 20 | CIFAR100 (vehicles 1) | 32 | 3000 |
| 21 | CIFAR100 (vehicles 2) | 32 | 3000 |
| 22 | aircraft | 64 | 6667 |
| 23 | cub | 64 | 10649 |
| 24 | dtd | 64 | 3760 |
| 25 | isic | 64 | 22802 |
| 26 | merced | 64 | 1890 |
| 27 | scenes | 64 | 14088 |
| 28 | ucf101 | 64 | 12024 |
| 29 | daimlerpedcls | 64 | 29400 |
| 30 | gtsrb | 64 | 26640 |
| 31 | kather | 64 | 4000 |
| 32 | omniglot | 64 | 25968 |
| 33 | svhn | 64 | 73257 |
| 34 | vgg-flowers | 64 | 2040 |
| 35 | bach | 64 | 320 |
| 36 | protein atlas | 64 | 12113 |
| 37 | minc | 64 | 51750 |
| 38 | ImageNet (bag) | 64 | 6519 |
| 39 | ImageNet (retriever) | 64 | 6668 |
| 40 | ImageNet (domestic cat) | 64 | 6750 |
| 41 | ImageNet (stick) | 64 | 6750 |
| 42 | ImageNet (turtle) | 64 | 6750 |
| 43 | ImageNet (finch) | 64 | 6750 |
| 44 | ImageNet (watchdog) | 64 | 6404 |
| 45 | ImageNet (footwear) | 64 | 6587 |
| 46 | ImageNet (salamander) | 64 | 6750 |
| 47 | ImageNet (anthropoid ape) | 64 | 6750 |
| 48 | ImageNet (elasmobranch) | 64 | 6750 |
| 49 | ImageNet (shorebird) | 64 | 6750 |
| 50 | ImageNet (housing) | 64 | 6605 |
| 51 | ImageNet (foodstuff) | 64 | 6750 |
| 52 | ImageNet (substance) | 64 | 6643 |
| 53 | ImageNet (spider) | 64 | 8100 |
| 54 | ImageNet (percussion instrument) | 64 | 8082 |
| 55 | ImageNet (New World monkey) | 64 | 8100 |
| 56 | ImageNet (big cat) | 64 | 8100 |
| 57 | ImageNet (box) | 64 | 7829 |
| 58 | ImageNet (fabric) | 64 | 7862 |
| 59 | ImageNet (kitchen appliance) | 64 | 7773 |
| 60 | ImageNet (mollusk) | 64 | 8100 |
| 61 | ImageNet (hand tool) | 64 | 8054 |
| 62 | ImageNet (butterfly) | 64 | 8100 |
| 63 | ImageNet (stringed instrument) | 64 | 8100 |
| 64 | ImageNet (boat) | 64 | 8006 |
| 65 | ImageNet (rodent) | 64 | 8006 |
| 66 | ImageNet (toiletry) | 64 | 7522 |
| 67 | ImageNet (computer) | 64 | 7696 |
| 68 | ImageNet (shop) | 64 | 9400 |
| 69 | ImageNet (musteline mammal) | 64 | 9450 |
| 70 | ImageNet (Old World monkey) | 64 | 9450 |
| 71 | ImageNet (bottle) | 64 | 9205 |
| 72 | ImageNet (fungus) | 64 | 9450 |
| 73 | ImageNet (truck) | 64 | 9309 |
| 74 | ImageNet (spaniel) | 64 | 9119 |
| 75 | ImageNet (sports equipment) | 64 | 9450 |
| 76 | ImageNet (game bird) | 64 | 9450 |
| 77 | ImageNet (seat) | 64 | 9126 |
| 78 | ImageNet (fruit) | 64 | 9450 |
| 79 | ImageNet (weapon) | 64 | 9450 |
| 80 | ImageNet (beetle) | 64 | 10800 |
| 81 | ImageNet (toy dog) | 64 | 9832 |
| 82 | ImageNet (decapod crustacean) | 64 | 10800 |
| 83 | ImageNet (fastener) | 64 | 10675 |
| 84 | ImageNet (timepiece) | 64 | 10164 |
| 85 | ImageNet (dish) | 64 | 10556 |
| 86 | ImageNet (mechanical device) | 64 | 10617 |
| 87 | ImageNet (colubrid snake) | 64 | 12150 |

Table 3: Format A (for activations and weights)

| index | # exponent bits | # mantissa Bits | total bit width |
|---|---|---|---|
| 1 | 3 | 1 | 5 |
| 2 | 3 | 2 | 6 |
| 3 | 3 | 3 | 7 |
| 4 | 3 | 4 | 8 |
| 5 | 4 | 1 | 6 |
| 6 | 4 | 2 | 7 |
| 7 | 4 | 3 | 8 |
| 8 | 4 | 4 | 9 |
| 9 | 5 | 1 | 7 |
| 10 | 5 | 2 | 8 |
| 11 | 5 | 3 | 9 |

Table 4: Format B (for optimizer)

| index | # exponent bits | # mantissa Bits | total bit width |
|---|---|---|---|
| 1 | 6 | 7 | 14 |
| 2 | 6 | 9 | 16 |
| 3 | 6 | 11 | 18 |
| 4 | 7 | 7 | 15 |
| 5 | 7 | 9 | 17 |
| 6 | 7 | 11 | 19 |
| 7 | 8 | 7 | 16 |
| 8 | 8 | 9 | 18 |
| 9 | 8 | 11 | 20 |

Table 5: Datasets and learning rates in Section 4.3

| index | dataset name | learning rate |
|---|---|---|
| 1 | CIFAR100 (aquatic mammals) | 0.01 |
| 2 | CIFAR100 (fish) | 0.01 |
| 3 | CIFAR100 (flowers) | 0.01 |
| 4 | CIFAR100 (food containers) | 0.01 |
| 5 | CIFAR100 (fruit and vegetables) | 0.01 |
| 6 | CIFAR100 (household electrical devices) | 0.01 |
| 7 | CIFAR100 (household furniture) | 0.01 |
| 8 | CIFAR100 (insects) | 0.01 |
| 9 | CIFAR100 (large carnivores) | 0.01 |
| 10 | CIFAR100 (large man-made outdoor things) | 0.01 |
| 11 | CIFAR100 (large natural outdoor scenes) | 0.01 |
| 12 | CIFAR100 (large omnivores and herbivores) | 0.01 |
| 13 | CIFAR100 (medium-sized mammals) | 0.01 |
| 14 | CIFAR100 (non-insect invertebrates) | 0.01 |
| 15 | CIFAR100 (people) | 0.01 |
| 16 | CIFAR100 (reptiles) | 0.01 |
| 17 | CIFAR100 (aquatic mammals) | 0.001 |
| 18 | CIFAR100 (fish) | 0.001 |
| 19 | CIFAR100 (flowers) | 0.001 |
| 20 | CIFAR100 (food containers) | 0.001 |
| 21 | CIFAR100 (fruit and vegetables) | 0.001 |
| 22 | CIFAR100 (household electrical devices) | 0.001 |
| 23 | CIFAR100 (household furniture) | 0.001 |
| 24 | CIFAR100 (insects) | 0.001 |
| 25 | CIFAR100 (large carnivores) | 0.001 |
| 26 | CIFAR100 (large man-made outdoor things) | 0.001 |
| 27 | CIFAR100 (large natural outdoor scenes) | 0.001 |
| 28 | CIFAR100 (large omnivores and herbivores) | 0.001 |
| 29 | CIFAR100 (medium-sized mammals) | 0.001 |
| 30 | CIFAR100 (non-insect invertebrates) | 0.001 |
| 31 | CIFAR100 (people) | 0.001 |
| 32 | CIFAR100 (reptiles) | 0.001 |
| 33 | CIFAR100 (aquatic mammals) | 0.0001 |
| 34 | CIFAR100 (fish) | 0.0001 |
| 35 | CIFAR100 (flowers) | 0.0001 |
| 36 | CIFAR100 (food containers) | 0.0001 |
| 37 | CIFAR100 (fruit and vegetables) | 0.0001 |
| 38 | CIFAR100 (household electrical devices) | 0.0001 |
| 39 | CIFAR100 (household furniture) | 0.0001 |
| 40 | CIFAR100 (insects) | 0.0001 |
| 41 | CIFAR100 (large carnivores) | 0.0001 |
| 42 | CIFAR100 (large man-made outdoor things) | 0.0001 |
| 43 | CIFAR100 (large natural outdoor scenes) | 0.0001 |
| 44 | CIFAR100 (large omnivores and herbivores) | 0.0001 |
| 45 | CIFAR100 (medium-sized mammals) | 0.0001 |

