# OpenReview forum: "How Low Can We Go: Trading Memory for Error in Low-Precision Training"
_ICLR.cc/2022/Conference — ICLR 2022 Poster_

### Official Review · Reviewer_ANon · 2021-10-20

**Correctness:** 3
**Technical Novelty And Significance:** 2
**Empirical Novelty And Significance:** 2
**Recommendation:** 5
**Confidence:** 3

**Main Review:**

1. Incomplete experimental results
1-1) There is a limitation that most of the experiments were performed on CIFAR-10. Also, experimental results on more networks (e.g., MobileNet, EfficientNet, more ResNets) should be included.
1-2) Although the proposed method considers both training and inference, performance comparison with existing SOTA studies for each phase should be performed.
1-3) It is necessary to show whether the proposed method is effective not only for classification but also for more practical detection and segmentation networks.

2. It is difficult to understand the actual use-case of the proposed method. Most of the experimental results also appear to support the feasibility of meta-training. Therefore, it is difficult to be sure how practical the proposed method will be. It would be good to think about this comment in relation to the comments about additional experiments (i.e., Comment 1).

3. It is difficult to agree with the argument of this paper because the motivation of this paper is not sufficiently presented. The motivation of this study needs to be clearly presented at the beginning to secure differentiation from many recently published low-precision training studies.

4. It is necessary not to just consider memory capacity as a resource, but to consider practical indicators such as power consumption or memory R/W portion as the target of trading.

5. Although the proposed method suggests optimization considering the memory budget, it is necessary to consider the computational power together. It is not practical to consider only the memory budget because various designs using memory reuse, etc. are possible depending on the computation method/capability.

6. It would be nice if comparative analysis and comparison results with custom hardware designs are presented.

**Summary Of The Paper:**

To achieve the goal of efficiently selecting the best low-precision configuration within the memory budget, this paper proposed Pareto Estimation to Pick the Perfect Precision (PEPPP) by using matrix factorization to find non-dominated configurations (the Pareto frontier) with a limited number of network evaluations. Although this paper presents a solution for low-precision training that has recently gained a lot of attention, there are some concerns as a result of my review of this paper. (Please refer to my main review)

**Summary Of The Review:**

The motivation and contribution need to be made clearer. In addition, the experiment part that supports the contribution needs to be thoroughly supplemented. Please address my concerns in "Main Review" through the rebuttal process.

---

> ### Author Response · Authors · 2021-11-19
> **Response to Reviewer ANon**
>
> Thank you for your careful reading and thoughtful reviews. Let us address your comments below.
>
> > Incomplete experimental results 1-1) There is a limitation that ...
>
> In terms of datasets, as mentioned in the first paragraph of Section 4 (on Page 6), we include 20 CIFAR-100 subsets, 50 ImageNet subsets and 16 datasets from the visual domain benchmark. Figure 1(a), 7 and 10 show CIFAR-10 results as a running example, but CIFAR-10 is only one of the 87 datasets we have in our experiments.
>
> In terms of architectures, While the main results with our paper are on ResNet-18 only, in Section 4.1 and Appendix E.5 we show additional results on more network architectures (ResNet32, VGG-11, VGG-13, VGG-16, and VGG-19). Meta-learning across architectures works better than considering each architecture separately. We have updated Section 4.4 to make it clearer.
>
> > 1-2) Although the proposed method ...
>
> As for the SOTA of the performance of individual models on individual datasets, as mentioned in the second from last paragraph of Section 1 (on Page 2), our goal is to reduce training or deployment costs of customized models trained on proprietary datasets , not pursuing SOTA models on ML benchmarks. In the first paragraph of Section 4.3 (on Page 9), we show our performance data generalizes to produce near-SOTA results.
>
> As for the SOTA on meta-training and meta-inference (meta-test), we are not aware of any previous work that picks the best low-precision configuration on a user-specified dataset. That said, if there is a SOTA you would like to suggest we should compare to, we would be happy to look into it.
>
> > 1-3) It is necessary to show ...
>
> We agree that it is important to investigate additional experimental domains, however we believe this is best suited for follow-up works. We would like to note that it is standard in the machine learning literature to demonstrate viability of methods on image classification tasks on CIFAR-10 and ImageNet, and this paper is already long enough.
>
> > It is difficult to understand the actual use-case ...
>
> Imagine a company with a tight budget, and still want to seek a model that has excellent performance on their proprietary dataset. Our method helps to find the best low-precision configuration that fits into the memory limit dictated by the budget. This is achieved by evaluating (training and testing) only 10-15% of all possible models.
>
> > It is difficult to agree with the argument of this paper because the motivation of this paper is not sufficiently presented ...
>
> Please see Section 1 for the motivation, and the second from last paragraph in Section 1 (on Page 2) for the relationship with recent low-precision training works. Please feel free to ask for further clarifications for anything ambiguous.
>
> > It is necessary not to just consider memory capacity as a resource ...
>
> Figure 9 shows we only need to measure 15% of all configurations in the meta-training phase to have a good enough Pareto frontier estimation; Figure 11 shows about 10% is needed for meta-test. This means with our approach, we only need 10-15% of the energy consumed by exhaustive search.
>
> > Although the proposed method suggests optimization considering the memory budget ...
>
> Memory can indeed be reused, but in large-scale hyperparameter tuning experiments, people often need to run a ton of experiments in parallel. In such cases, a reduction in total memory usage would result in a reduction in compute.
>
> Also, we would like to point out that our approach is not limited to studying the error-memory tradeoff. Future works on directly studying the memory-compute tradeoff (with compute coming from direct measurements on custom hardware, or empirical performance models) would also be worthwhile.
>
> > It would be nice if comparative analysis and comparison results with custom hardware designs are presented.
>
> Absolutely. However, this paper is already unfortunately complicated (and long) enough.

---

### Official Review · Reviewer_HPBj · 2021-11-02

**Correctness:** 4
**Technical Novelty And Significance:** 4
**Empirical Novelty And Significance:** 4
**Recommendation:** 6
**Confidence:** 3

**Main Review:**

The paper formulates the determination of low-precision configuration as a multi-objective optimization. The overall objective is to allow ML engineers to select optimal precision that lets them train the desired model with less energy.

The key idea seems to be using meta-learning and testing to determine these Pareto frontiers. The overall methodology is sound and is evaluated thoroughly. However, one limitation is that it doesn't provide evaluation on ImageNet which usually is a good reference point for the readers.

I am not an expert in low-precision training. However, I like the overall idea where they formulated the configuration determination into a multi-objective optimization setting.

Questions:
* Are there results for energy reduction?

**Summary Of The Paper:**

The paper aims to find Pareto frontiers for the precision given memory budget. The overall objective is to allow ML engineers to select optimal precision that lets them train the desired model with less energy. To do this, the paper formulates the problem into multi-objective optimization problem and use meta-learning to minimize the number of low-precision training data points for this optimization. Then, the paper uses the term "meta-test" that refers to transferring the information from known tasks to estimate Pareto frontiers of low-precision training on new unseen tasks.

**Summary Of The Review:**

I believe the paper proposes interesting approach in selecting low-precision training configurations. Formulating into multi-objective optimization and using meta-training/testing to find Pareto frontier of configurations for the overall accuracy seems like the key contribution of the paper. I am not an expert on low-precision training, but to the best of my knowledge, the overall approach seems reasonable and the evaluation seems okay. However, I believe it would provide better reference for the readers if the authors provide evaluation on ImageNet.

---

> ### Author Response · Authors · 2021-11-20
> **Response to Reviewer HPBj**
>
> Thank you for your careful reading and thoughtful reviews. Let us address your comments below.
>
> > However, one limitation is that it doesn't provide evaluation on ImageNet which usually is a good reference point for the readers.
>
> Among all 87 datasets we evaluate on, there are 50 ImageNet subsets (listed in Page 20 Table 2). Throughout experimental results, we show our method can effectively learn the error-memory trade-offs across datasets, including these ImageNet subsets that account for a majority of all datasets.
>
> That said, we also studied the performance of PEPPP on the entire ImageNet: using all but the 50 ImageNet subsets among 87 datasets (namely the first 37 datasets listed in Page 20 Table 2) for meta-training, and using the entire ImageNet for meta-test. On the entire ImageNet, we train ResNet-18 with each of the 99 low-precision configurations and the same hyperparameter settings as described in the second paragraph of Section 4 (at the bottom of Page 6), except that we use batch size 2048 and learning rate 0.064. Across the range of the relative memory usage in the x axis of Figure 11, ED-MF (PEPPP) consistently ranks among top 3 of the 5 methods, and is top 2 (and often top 1) at small memory usage (below 5% of exhaustive search).
>
> > Are there results for energy reduction?
>
> Figure 9 shows we only need to measure 15% of all configurations in the meta-training phase to have a good enough Pareto frontier estimation; Figure 11 shows a similar ratio is needed for meta-test. This means with our approach, we only need 10-15% of the energy consumed by exhaustive search.

---

> > ### Comment · Reviewer_HPBj · 2021-11-22
> > **Thank you**
> >
> > I thank the authors for their detailed response.
> > I will stay with 6.

---

### Official Review · Reviewer_ezUH · 2021-11-02

**Correctness:** 3
**Technical Novelty And Significance:** 3
**Empirical Novelty And Significance:** 3
**Recommendation:** 6
**Confidence:** 4

**Main Review:**

The authors propose an interesting, one-of-a-kind two-step approach for error-memory trade-off in low-precision training/inference of neural networks. I appreciate the design of the method, the utility of the method (I find it very useful for the community working in the field) as well as the extensive set of measurements. The paper however could be improved especially in terms of clarity, especially in the experimental section. Please find below a few questions below:

- Figure 1(b): it is not clear what the numbers in the parenthesis are. Memory requirements?
- Is the meta-training performed per neural network architecture? This is my understanding, and if this is so, please clearly specify this in the description of the method (e.g., in Figure 2).
- Was regularization used in the linear regression model? If so, how sensitive is the estimated Pareto frontier to the regularization parameter?
- Configuration formats A and B (Section 2): the weights should also be mentioned, as you are using one of the two formats for weights as well, not only for the optimizer and activation.
- How is the memory computed? Maximum memory measured at train/inference time? Or theoretical computation knowing the neural network architecture? My understanding is the latter, as you compute the full memory matrices both at train and test times. It should be clearly mentioned in the paper.
- Figure 6b deserves more discussion. How it was generated, what it means. This is important as it supports one of the main properties of the method: the rapid decaying of the singular values of the error matrix. On a related note, how to choose the dateset-configuration data-points to guarantee this property?
- Please define the propensity score.
- In section 3.2, please define all the notations (e.g., yi).
- In section 4.2 how was rank 3 chosen?
- Figure 11: it is not clear what the x axis represents. For setting I, there is no memory cap, thus what does relative memory usage mean?
- In Section 4, the authors mention different neural network architectures: ResNet-18, ResNet-34 and VGG. For which architecture are the results shown in Section 4? If for all of them, how are the performance metrics aggregated across architectures? Averaged across architectures? This is unfortunately not clear.

**Summary Of The Paper:**

The paper proposes a two-step methodology for evaluating the error-memory trade-off when employing low-precision training/inference in neural networks. Although low-precision training/inference in deep learning is a widely studied topic, this paper seems to be the first one to study this error-memory tradeoff without exhaustive search.

The proposed idea is inspired from the multi-objective optimization field. The authors propose to identify the Pareto frontier that would allow a user to identify the optimal precision (lowest error) given a certain memory constraint. The proposed system has two phases: training and testing.

At training time, given a set of datasets, a network architecture and a set of low-precision configurations, the method samples dataset-configuration pairs to train and computes the misclassification error for each selected pair. Matrix factorization is used to compute low-dimensional embeddings for each configuration.

At test time, a subset of configurations is chosen for evaluation (using Design Experiments with Matrix Factorization) and the computed errors are used as input together with the configuration embeddings to a linear regression model in order to estimate the errors on the non-evaluated configurations. For both training and testing, the memory matrix is fully computed based on the network architecture and the low-precision format.

The method is evaluated in terms of convergence and Hypervolume difference of the true and estimated Pareto frontier and it outperforms in most of the cases other methods such a Bayesian Optimization or Random Selection with Matrix Factorization.


**Summary Of The Review:**

Although there are some items that should be better explained / clarified, overall I enjoyed reading this paper and find the addressed topic of high interest for the community working in the field of low-precision training/inference of neural networks. Although I am not familiar with the prior art, the paper seems to be the first to propose a method for analyzing the error-memory trade-off in a deep learning low-precision training/inference scenario without exhaustive search.

I like the two-step meta-training/meta-test approach, however I am not yet fully convinced about the main property of the meta-training algorithm, namely the rapid decaying of the singular values of the error matrix. In terms of evaluation, I appreciate the authors analyzing the performance of the meta-training approach using multiple metrics and providing recommendations for the optimal sampling conditions.

Also for the meta-test phase, the authors analyze the performance of the ED-MF approach to other methods such as bayesian optimization, random selection and QR with matrix factorization. ED-MF out-performs these methods in most of the cases. Some results are not clear unfortunately - see my questions in the main review above. Hopefully the results will become more clear during the paper discussion / rebuttal phase.

---

> ### Author Response · Authors · 2021-11-19
> **Response to Reviewer ezUH**
>
> Thank you for your careful reading and thoughtful reviews. Let us address your comments below.
>
> > Figure 1(b): it is not clear what the numbers in the parenthesis are ...
>
> Thanks for pointing out the ambiguity. At different numbers of activation bits, these numbers in parenthesis are the respective numbers of non-dominated configurations on all 87 datasets: there are 722 non-dominated configurations in total, 8.2 on average for each dataset (like the blue circles in Figure 1(a)). For example, with 5 activation bits (the green slice), 26% = 189/722.
>
> > Is the meta-training performed per neural network architecture? ...
>
> The meta-training can be performed either per architecture (before Section 4.4) or across architectures (in Section 4.4). This is not fundamental to our methodology, but rather a matter of how we do experiments. We have made it clearer in Section 4.4. Thanks for pointing it out.
>
> > Was regularization used in the linear regression model? ...
>
> Regularization was not used when the meta-training error matrix does not have missing entries (i.e., when we know the performance of every configuration on every meta-training dataset). We just do PCA to get the dataset and configuration embeddings. When we fill in the missing entries in a partially-observed error matrix, the $\lambda$ with a decaying schedule in the SoftImpute algorithm (Page 13 Appendix B Algorithm 1) can be viewed as a nuclear norm regularization. We tune the initial value, and use a popular decaying schedule that comes from TFOCS backtracking: this is an algorithmic decision we did not further fine-tune. Then on the imputed matrix, we do PCA with no regularization to get the embeddings.
>
> Here are some more details on tuning $\lambda$ in SoftImpute: at each combination of the observation ratios {0.1, 0.15, 0.2, 0.25, 0.3, 0.35, 0.4, 0.45, 0.5} and $\lambda$ values {1e-4, 5e-4, 1e-3, 5e-3, 1e-2, 5e-2, 1e-1, 5e-1, 1, 5}, we check the relative error of the imputed meta-learning error matrix. $\lambda=0.1$ consistently achieves the smallest relative error at all the above observation ratios.
>
> > Configuration formats A and B: the weights should also be mentioned ...
>
> We used “weights” and “parameters” interchangeably in the original draft. In the revision, we have updated the terminology to only use “weights”. Thanks for pointing out the ambiguity!
>
> > How is the memory computed? ...
>
> The memory we use in experiments is the theoretical value at training time: how much memory we need to train the neural network architecture with a given optimizer. This means weights, activations and the optimizer are all taken into account (see Figure 5). We have made it clearer in Section 3.1.
>
> > Figure 6b deserves more discussion ...
>
> We described how Figure 6(b) is generated and its implication in the paragraph besides Figure 5 (the second paragraph in Page 5), and have made the statement clearer. Empirically, the rapid singular value decay was observed to help in meta-learning [1]; theoretically in an asymptotic manner, such a property does not require a specific selection of datasets and configurations to hold [2]. It is definitely worthwhile to explore when this property holds for a small or medium number of dataset-configuration data points, and that fits better in a future work.
>
> [1] Chengrun Yang, Yuji Akimoto, Dae Won Kim, Madeleine Udell. OBOE: Collaborative filtering for AutoML model selection. KDD 2019.
>
> [2] Madeleine Udell, Alex Townsend. Why are big data matrices approximately low rank?. SIAM Journal on Mathematics of Data Science, 2019.
>
> > Please define the propensity score.
>
> Thanks for pointing it out. We have replaced the terminology with “sampling probability”.
>
> > In section 3.2, please define all the notations (e.g., yi).
>
> Thanks for catching it. We have added the definition to Section 2 and Section 3.2.
>
> > In section 4.2 how was rank 3 chosen?
>
> We choose rank 3 because it is the smallest number of principal components of the meta-training error matrix that can explain enough variance of the data in the matrix. In Figure 6(b), this rank corresponds to the index i right after the initial rapid singular value decay. To make it clearer, we have added the explained variance plot that corresponds to Figure 6(b) as Appendix E Figure 14, and added a paragraph to explain it at the beginning of Appendix E.
>
> > Figure 11: it is not clear what the x axis represents ...
>
> The x axis is the memory usage relative to exhaustively searching all permissible configurations (in the title of Figure 11). When there is no memory cap, this means the x axis is the memory usage relative to exhaustively evaluating all 99 configurations listed in Appendix A (Page 13), Table 3 and 4 (Page 20).
>
> > ... For which architecture are the results shown in Section 4? ...
>
> The results shown in the main paper are on ResNet-18 only, and we show in Appendix E.5 that the ResNet-18 results are generalizable to other architectures. We have updated Section 4.4 to make it clearer.

---

> > ### Comment · Reviewer_ezUH · 2021-11-29
> > **Reply to authors**
> >
> > Thank you for the additional comments and clarifications. I appreciate the time the authors and the reviewers spent to discuss this manuscript. I will keep my current score.

---

### Official Review · Reviewer_M7yh · 2021-11-02

**Correctness:** 4
**Technical Novelty And Significance:** 3
**Empirical Novelty And Significance:** 3
**Recommendation:** 8
**Confidence:** 4

**Main Review:**

Strengths:
1- The proposed approach is evaluated in large and variant benchmarks which means the proposed approach can be generalized to various datasets and models
2- The paper is well written, and the algorithms and methods are clearly explained.

Weaknesses:
1- The quality of the proposed approach based on convergence metric and hypervolume difference is not obvious.
2- The overhead of meta learning approach in term of time needs to be reported ( how many hours it takes to select precision configuration )
3- I would recommended to compare the meta learning approach with other multi objective approaches and report the overhead of each method.
4- The related works such as [1,2] on selecting trade-off between precision and others metric is not discussed in the paper.
[1] Hashemi, Soheil, et al. "Understanding the impact of precision quantization on the accuracy and energy of neural networks." Design, Automation & Test in Europe Conference & Exhibition (DATE), 2017. IEEE, 2017.
[2]  Langroudi, Hamed F., et al. "Cheetah: Mixed low-precision hardware & software co-design framework for dnns on the edge." arXiv preprint arXiv:1908.02386 (2019).

**Summary Of The Paper:**

The paper proposes a meta learning approach to select the best trade-of between memory usage and bit precision for DNN training. This approach is evaluated on 87 datasets and 99 bit-precision configurations. The result shows that in general 6 bit floats (1,4,1) for activations and weights and 14 bit floats (1,6,7) for optimizer parameters is the best choice in term of memory and accuracy trade-off.

**Summary Of The Review:**

As a reviewer mentioned, the idea behand the paper is interesting. However, there are couple of comments needs to be addressed. I recommend this paper marginally accepted.

---

> ### Author Response · Authors · 2021-11-19
> **Response to Reviewer M7yh**
>
> Thank you for your careful reading and thoughtful reviews. Let us address your comments below.
>
> > The quality of the proposed approach based on convergence metric and hypervolume difference is not obvious.
>
> To quantify the quality of Pareto frontier estimates, convergence and hypervolume difference are the most intuitive metrics we can come up with and can find. We would definitely appreciate suggestions on more intuitive metrics.
>
> > The overhead of meta learning approach in term of time needs to be reported ...
>
> It takes ~3 minutes to make one measurement (namely, evaluating one configuration on one dataset) on an ImageNet or CIFAR-100 subset in our setting. In meta-training, Figure 9 and 17 show that we need to make only 15% of all possible measurements to get a good-enough estimate of the Pareto frontiers on meta-training datasets, which would take ~86 GPU hours. In meta-test with ED-MF, we only need to make ~3 measurements to get a good-enough estimate of the Pareto frontier on the meta-test dataset. This takes only ~10 minutes.
>
> Note that the intended use case for practitioners is meta-test: to take advantage of the information from previously run experiments. We can think of the time spent in meta-training as aggregating across many users.
>
> > I would recommended to compare the meta learning approach with other multi objective approaches and report the overhead of each method.
>
> We would love to, but the issue is that we do not have parametric forms of the two costs that typically exist in multi-objective optimization problems. Building parametric cost models would incur additional complexities and suits better for a future work. That said, if there is a particular multi-objective approach you would like to suggest we should compare to, we would be happy to look into it.
>
> > The related works such as ... on selecting trade-off between precision and others metric is not discussed in the paper ...
>
> Thanks for the pointers. We have added these two works to our discussion in the revised version. Please see the bottom of Page 2.

---

> > ### Comment · Reviewer_M7yh · 2021-11-28
> > **Post Rebuttal**
> >
> > I appreciate the authors' responses to my comments. After carefully reading your responses and other reviewers’ comments, I have decided to increase my score to 8.

---

### Decision · Program_Chairs · 2022-01-20

**Decision:**

Accept (Poster)

**Comment:**

The paper considers a learning problem to determine the best low-precision configuration within the memory budget.  It is an interesting problem that could be of interest to the community.  Overall, the reviewers were fairly positive on the paper and believe the paper give interesting insights into how to use limited memory for learning.